# Serum EphA2 as a Promising Biomarker for the Early Detection and Diagnosis of Colorectal Cancer

**DOI:** 10.3390/biom14121504

**Published:** 2024-11-26

**Authors:** Shunsuke Sakuraba, Akihiro Koizumi, Takumi Iwasawa, Tomoaki Ito, Kazunori Kato

**Affiliations:** 1Department of Surgery, Juntendo University Medical School, Shizuoka Hospital, Shizuoka 410-2295, Japan; lunlun8shunsuke@gmail.com (S.S.); tomo-ito@juntendo.ac.jp (T.I.); 2Shizuoka Medical Research Center for Disaster, Shizuoka 410-2295, Japan; iwasawa@toyo.jp; 3Institute of Life Innovation Studies, Toyo University, Tokyo 115-8650, Japan; 4Department of Nutrition Sciences, Graduate School of Health and Sports Sciences, Toyo University, Tokyo 115-8650, Japan

**Keywords:** EphA2, serum EphA2, biomarker, colorectal cancer

## Abstract

Background: EphA2, a receptor-type tyrosine kinase, is overexpressed in several cancers, including colorectal cancer (CRC), and can be detected as soluble EphA2 in serum. This study aimed to investigate the relationship between soluble EphA2 and CRC. Methods: Serum samples were collected from 65 patients with CRC and 19 healthy individuals. Time-series changes in soluble EphA2 levels were measured in CRC cell lines to verify the release of EphA2 into the culture medium. Results: Soluble EphA2 levels were significantly higher in patients than in healthy individuals (*p* < 0.0001). Specifically, even in early-stage cancer, there was a notable difference between healthy individuals and patients with Stage I CRC (*p* = 0.00298), highlighting the potential of EphA2 as a biomarker for early detection. Additionally, correlations were observed with tumor size (*p* = 0.0346), depth of invasion (*p* = 0.0311), and lymphatic invasion (*p* = 0.0431). A receiver operating characteristic (ROC) analysis yielded an area under the curve (AUC) of 0.90 with 93.8% sensitivity and 78.9% specificity at a cutoff value of 448 pg/mL. Conclusions: These findings suggest that serum EphA2 could serve as a valuable biomarker for the early detection of CRC, offering a practical and minimally invasive alternative to conventional tumor markers.

## 1. Introduction

Worldwide, the number of new colorectal cancer (CRC) cases is increasing annually. In 2022, more than 1.9 million new cases of colorectal cancer and more than 904,000 deaths were reported, accounting for approximately 1 in 10 cancer cases and deaths [1]. In recent years, increased morbidity and mortality have also been noted in adults younger than 50 years of age with the incidence of CRC in the United States increasing by 22% from 2000 to 2013 and mortality increasing by 13% from 2000 to 2014 [2,3]. New screening methods are expected to lead to earlier detection and treatment, reducing mortality rates.

EphA2 is a member of the erythropoietin-producing hepatocellular (Eph) receptor family. Eph receptors are receptor-type tyrosine kinases that reside in the plasma membrane and form the largest family of receptor-type tyrosine kinases. They are classified into two subclasses (EphA and EphB) based on their amino acid sequences and affinity for their ligands [4]. Nine EphA and five EphB subclasses have been identified in humans [5]. EphA2 is mainly restricted to proliferating epithelial cells in adults but is also overexpressed in cancer cells. For example, EphA2 is abundantly expressed in prostate [6], lung [7], esophageal [8], colon [9], cervical [10], ovarian [11], breast [12], and skin cancer [13].

Although many previous studies have reported the expression of EphA2 in harvested cancer tissues, it is released from the cell surface by enzymes and detected as soluble EphA2 in the serum [6,7,8,10,11]. Expanding research on the measurement of serum EphA2 and its significance could pave the way for the real-time monitoring of cancer progression and further utilization as a therapeutic target in the future [14,15]. Cancer cells are thought to undergo changes in malignancy at various stages of progression and acquire or lose functions, such as developing drug resistance, during pharmacological treatment [16,17]. However, real-time tissue sampling at different stages of cancer treatment is rarely feasible. Advancing research on the measurement of biomarkers released from cancer cells into the serum, such as through liquid biopsy, is considered valuable and holds significant potential for improving cancer management. If soluble EphA2 levels in the serum can be used in colorectal cancer patients, it could provide a more convenient and less invasive method for their diagnosis and treatment. Traditionally, screening for colorectal cancer has relied on methods such as fecal occult blood tests, colonoscopy, and imaging techniques, which, while effective, can be highly expensive, invasive, and unable to screen many patients at once, limiting their widespread use in routine screening [18,19]. The introduction of a novel biomarker for the early detection of colorectal cancer, detectable through an ELISA-based assay, presents significant financial implications both in terms of cost and cost-effectiveness. The ELISA method, leveraging our newly identified biomarker, provides a less invasive and potentially lower-cost alternative. The test’s ability to be performed in a standard laboratory setting without the need for specialized equipment further enhances its cost-effectiveness. Moreover, the simplicity and non-invasive nature of a blood test can significantly increase patient compliance and screening uptake, as it is easier to administer across various healthcare settings and is much more acceptable to patients, potentially leading to higher detection rates and earlier interventions.

Based on these facts, this study aimed to measure time-series changes in soluble EphA2 released from CRC cell lines and to measure soluble EphA2 in serum samples from patients with CRC to examine the relationship between EphA2 and the clinical characteristics of these patients.

## 2. Materials and Methods

### 2.1. Cell Lines and Cell Culture

HCT-116 (colorectal adenocarcinoma) cells were purchased from ATCC (Manassas, VA, USA). HUVEC (vascular endothelial cells derived from the umbilical vein) were purchased from KURABO (Osaka, Japan). Cells were cultured in RPMI1640 (Nacalai Tesque, Kyoto, Japan) supplemented with 10% fetal bovine serum (Biosera, Cholet, France) and antibiotics (100 U/mL penicillin and 100 μg/mL streptomycin) at 37 °C in a humidified atmosphere containing 5% CO_2_.

### 2.2. Patient Enrollment and Sample Collection

From March 2014 to July 2015, 65 patients with CRC who underwent surgery at Juntendo University Shizuoka Hospital were enrolled in this study. Upon admission, 12 mL of peripheral venous blood was collected along with a preoperative blood test. Blood samples were centrifuged at 3000× *g* for 15 min, and the upper layer of serum was transferred into a 1.5 mL EP tube. The collected samples were stored at −80 °C until the analysis.

In addition, 19 healthy human serum samples were collected for comparison with samples from CRC patients and stored using similar processing methods. The healthy donors included 10 males and 9 females of 30–50 years of age with no history of cancer.

Pathological data, such as tumor size, lymph node metastasis, tumor sites, tumor stage, and laboratory serological test results, such as carcinoembryonic antigen (CEA) and carbohydrateantigen19-9 (CA19-9), were examined in the CRC group. The 8th edition of colorectal cancer staging criteria (UICC) was used in this study. The tumor site was divided into right and left sides based on the proximal two thirds of the transverse colon.

### 2.3. Monoclonal Antibodies to Human EphA2

Hybridomas producing mAbs against human EphA2 were generated by immunizing BALB/c mice with human EphA2 proteins purified from the supernatants of HEK239-F transfectants expressing human EphA2. The immunized splenocytes were then fused to P3 × 63Ag8U mouse myeloma cells. The supernatants were harvested 14 days post-fusion and screened for reactivity against EphA2 transfectants by an ELISA. After cloning by limiting dilution twice, we selected hybridoma cell colonies (mouse IgG1 and κ). To confirm this, the reactivity of mAbs to EphA2 transfectants was examined using flow cytometry (FACSCalibur, BD Immunocytometry Systems, Franklin Lakes, NJ, USA).

### 2.4. In-House ELISA for Detecting of Human EphA2

An in-house ELISA was developed to detect the serum EphA2 levels. A 96-well EIA/RIA plate (Corning, Steuben County, NY, USA) was coated with 5.0 µg/mL of the captured (human anti-EphA2) antibody. After the plate was kept overnight at 4 °C, it was blocked with SuperBlock Blocking Buffer (Thermo Fisher Scientific, Waltham, MA, USA) overnight at 4 °C. Serum samples (100 μL) or recombinant human EphA2 protein (R&D Systems, Minneapolis, MN, USA) diluted with Can Get Signal Solution 1 (TOYOBO, Osaka, Japan) to a final dilution of 1:5 were added in duplicate in each well. After incubation at room temperature for 2 h, the wells of the plate were washed three times with washing buffer (PBS-Tween 20 (0.05%); Merck, Darmstadt, Germany). Thereafter, 100 μL of diluted biotinylated detection antibody in Can Get Signal Solution 2 (TOYOBO) (to a final concentration of 0.13 μg/mL) was added to each well, and the plate was incubated at room temperature for 1 h. The wells were then washed four times. Then, 100 μL/well of diluted streptavidin poly horseradish peroxidase (Thermo Fisher Scientific, Waltham, MA, USA) at a final dilution of 1:1000 in ELISA Assay Diluent (BioLegend, San Diego, CA, USA) was added for 45 min at room temperature. The wells were then washed five times. TMB 1-component microwell peroxidase substrate, SureBlue (SeraCare Life Sciences Inc., Milford, MA, USA) 100 μL/well, was added for 15 min at room temperature. To stop the reaction, 100 μL/well sulfuric acid was added. The absorbance of each well was measured at a wavelength of 450 nm and a reference wavelength of 570 nm using an automated microplate reader (SpectraMax iD3, Molecular Devices, San Jose, CA, USA) with a compatible software program (SoftMax Touch (version 1.2.0.0), Molecular Devices, San Jose, CA, USA).

### 2.5. Western Blotting

Cells were cultured in a 10 cm dish (Iwaki Co., Tokyo, Japan). Cells were washed twice with PBS and then lysed with RIPA buffer (25 mM Tris, 150 mM KCl, 5 mM ethylenediaminetetraacetic acid (EDTA), 50 mM Na_3_VO_4_, 1% NP40 nonionic detergent, 0.5% sodium deoxycholate solid, 0.1% sodium dodecyl sulfate (SDS), and 1% protease inhibitor cocktail (Sigma-Aldrich, Shinagawa, Tokyo, Japan, pH 7.5)). The total protein (40 µg) was separated by electrophoresis on a 4–12% gradient polyacrylamide gel (Bio-Rad Laboratories, Hercules, CA, USA) using Tris-glycine sodium dodecyl sulfate (SDS) buffer (25 mM Tris, 192 mM glycine, 0.1% SDS) (SDS-PAGE). The separated proteins were transferred to polyvinylidene difluoride membranes (Thermo Fisher Scientific Inc., Waltham, MA, USA) for immunoblotting. Primary antibodies were used at the following dilutions and incubated overnight at 4 °C with anti-EphA2 (230-1, 1:1000) and anti-β-actin (3700T, 1:1000) antibodies from Cell Signaling Technology (Danvers, MA, USA). Secondary antibodies, anti-rabbit IgG horseradish peroxidase (HRP)-linked antibodies (18-8816-31) and anti-mouse IgG HRP-linked antibodies (18-8817-31) from Rockland Immunochemicals, Inc. (Limerick, Montgomery County, PA, USA) were used at 1:2000 dilution for EphA2 and β-actin. β-actin was used for normalization. Immunoreactivity was visualized by chemiluminescence using an enhanced substrate for HRP detection (iBright CL1500; Thermo Fisher Scientific Inc., Waltham, MA, USA).

### 2.6. Flow Cytometry Analysis

Flow cytometry (BD FACSCalibur, Canton, MA, USA) was used to determine the epha2 expression in hct-116 cells and HUVEC cells. The flowjo software (version 10.10.0), program (bd biosciences) was used to process the data. Cultured cells were digested into single-cell suspensions by trypsin/collagenase and incubated with anti-epha2 antibody 230-1 (prepared by our group) or 57-1(prepared by our group), or isotype igg (BioLegend, San Diego, CA, USA) for 45 min on ice. After one wash, they were infiltrated with FITC-conjugated goat anti-mouse IGG secondary antibodies (biolegend) for 30 min. All samples were washed three times before detection by flow cytometry. For quantification, the median fluorescence signal was used to calculate the relative ratio to the fluorescence signal of the isotype control.

### 2.7. Statistical Analysis

Excel 2021 (Microsoft Corporation, Redmond, WA, USA) was used to compile clinical data. Spearman’s rank correlation coefficient was used to analyze the relationship between continuous variables, whereas the Mann–Whitney U test was used to analyze the relationship between continuous and categorical variables. In cases where multiple group comparisons were necessary, Steel’s test was applied as a non-parametric method to compare each group with a control. All *p*-values were two-sided, and *p*-values of ≤0.05 were considered statistically significant. All statistical analyses were performed using EZR version 1.68 (Saitama Medical Center, Jichi Medical University; http://www.jichi.ac.jp/saitama-sct/SaitamaHP.files/statmedEN.html (accessed on 19 November 2024); Kanda, 2012), which is a graphical user interface for R (The R Foundation for Statistical Computing, Vienna, Austria, Version 2.13.0). More precisely, it is a modified version of R Commander (version 1.6-3) designed to add statistical functions frequently used in biostatistics.

## 3. Results

### 3.1. EphA2 Is Strongly Expressed on the Surface of CRC Cells and Is Released from CRC Cells

To verify that EphA2 expressed on the cell surface is released by enzymes into the serum, the following experiments were performed. Colon cancer cell lines (HCT-116) and a normal cell line (HUVEC) were used. For these cell lines, the amount of EphA2 in the cell culture medium was measured on days 1, 2, 3, 4, and 7 from the start of the cell culture. In the culture medium of colon cancer cell lines, there was an 8.96-fold increase in EphA2 levels in HCT-116 cells from days 1 to 7 (Figure 1A). The normal cell line also showed an increase, but the rate of increase was 2.98-fold from day 1 to day 7, which was less than that in the colon cancer cell lines (Figure 1A). When the expression of EphA2 in cells was evaluated, the EphA2 expression of colon cancer cells was greater than that of HUVECs (Figure 1B,C). The cell surface expression was also higher in HCT-116 cells than in HUVECs (Figure 1D,E).

### 3.2. Soluble EphA2 in Serum Samples from CRC Patients and Its Clinical Correlations

Table 1 shows the background characteristics of CRC patients.

Soluble EphA2 levels were compared between healthy individuals and CRC patients. The box plot shows the distribution of EphA2 levels in the two groups (Figure 2A). A Mann–Whitney U test revealed a significant increase in EphA2 levels between healthy individuals and patients with CRC (*p* < 0.0001). A comparison of the EphA2 levels between the healthy group and each stage (I–IV) of cancer patients revealed that the EphA2 levels were significantly higher in cancer patients (healthy individuals vs. Stages I, II, III, IV; *p* = 0.00298, *p* = 0.00000734, *p* = 0.000254, *p* = 0.000542, respectively) (Figure 2B).

We calculated the cutoff value for serum EphA2 in healthy individuals and patients with colorectal cancer. In this analysis, the area under the curve (AUC) was 0.90, indicating a high discriminative ability. The optimal cutoff value was 448 with 93.8% sensitivity and 78.9% specificity based on this threshold (Figure 3).

Table 2 presents the serum EphA2 levels (pg/mL) in CRC patients categorized by various clinical characteristics, including tumor location, tumor size, depth of invasion, lymph node metastasis, distant metastasis, vascular invasion, lymphatic invasion, pathological stage, CEA levels, and CA19-9 levels (Table 2). Significant differences in serum EphA2 levels were observed in patients according to tumor size (*p* = 0.0346), depth of invasion (*p* = 0.0311), and lymphatic invasion status (*p* = 0.0431). The other clinical characteristics had no significant association with the serum EphA2 levels.

## 4. Discussion

EphA2, which is cleaved in the extracellular region by membrane type 1 matrix metalloproteinase (MT1-MMP) and released extracellularly, is not highly enzymatically active; rather, colon cancer expresses high levels of EphA2 [20,21]. Therefore, measuring EphA2 released into the bloodstream from the surface of cancer cells might allow for the direct observation of changes in the expression of EphA2 in tumor cells over time. In this study, we confirmed the feasibility of measuring EphA2 released into the extracellular space over time using colorectal cancer cell lines. We used the HCT-116 colorectal cancer cell line, as it was the only one available for this investigation. Martini et al. have also investigated the expression of EphA2 in other colorectal cancer cell lines, including SW620, LOVO, SW480, and HCT15, as well as in cell lines resistant to the anti-EGFR antibody cetuximab, such as GEO-CR and SW48-CR, and in cetuximab-sensitive cell lines, GEO and SW48. EphA2 expression was consistently observed across all these colorectal cancer cell lines [9]. It is therefore speculated that through the same mechanisms described above, overexpressed EphA2 could be detectable as serum EphA2. These findings suggest that once the significance of EphA2 measurement in colorectal cancer is further elucidated, serum EphA2 could potentially be used for the early detection and monitoring of colorectal cancer as a liquid biopsy.

In a comparative analysis with healthy individuals, serum EphA2 levels were significantly higher in patients with CRC than in the healthy control group. Existing biomarkers for colorectal cancer, such as CEA and CA19-9, are widely used for recurrence diagnosis and disease assessment [22,23]. However, the use of these biomarkers as screening tests is currently not recommended, and there are currently no blood test markers available for the early detection of colorectal cancer [24]. In our current data as well, out of 65 colorectal cancer patients, 25 (38.5%) had CEA levels above the standard value of 5 ng/mL, and 10 (15.4%) had CA19-9 levels exceeding 37 U/mL. We found that serum EphA2 levels were significantly elevated in patients with Stage I colorectal cancer compared to healthy individuals. This observation suggests that EphA2 could be particularly useful as a biomarker for the screening and early diagnosis of colorectal cancer. While further research is required to fully determine its role in later stages and its potential impact on prognosis, our findings currently support its primary utility in early detection efforts. The early detection of elevated EphA2 levels may enable timely intervention, potentially improving outcomes for colorectal cancer patients.

In this study, when the cutoff value for EphA2 was determined for CRC patients and healthy individuals, the optimal value was 448. This cutoff value achieved 93.8% sensitivity, 78.9% specificity, and an area under the curve (AUC) value of 0.903, suggesting that EphA2 has potential as a useful biomarker for screening tests.

In the comparison between the histopathological data of colorectal cancer and serum EphA2 levels, higher EphA2 levels were observed to be associated with a larger tumor size and greater depth of invasion. This suggests that more EphA2 is released from the cell surface as the tumor size increases. Moreover, as the depth of invasion increases and the tumor progresses into the submucosa, EphA2 is more likely to be released into the bloodstream. In contrast, no significant correlation was found between serum EphA2 levels and clinical parameters, such as lymph node metastasis, distant metastasis, and clinical stage. Previous studies have also shown inconsistent findings regarding the correlation between clinical stage or prognosis and serum EphA2 levels. Thus, this relationship remains unclear [25,26,27]. These contradictory results may be related to the complex properties and varied roles of the EphA2 molecule (both the receptor and ligand). For example, EphA2, which functions as a receptor, is implicated in the control of opposing signals that drive carcinogenesis. In contrast to the cancer-promoting epidermal growth factor receptor (EGFR), EphA2 exerts tumor-suppressive effects through the ligand-dependent tyrosine phosphorylation (pY588) of specific tyrosine residues in the presence of its ligand. Conversely, in the absence of the ligand, EphA2 collaborates with downstream signals of EGFR, inducing the phosphorylation of serine residues (pS897) and promoting carcinogenesis [28]. It may also be influenced by the sample size and EphA2 detection methods.

In the future, EphA2 expression has the potential to be applied for the treatment of colorectal cancer. Additionally, the EphA2 expression may be useful for monitoring cancer during postoperative surveillance if it is effective in distinguishing between healthy individuals and patients with cancer. Currently, inhibitors targeting the epidermal growth factor receptor (EGFR) are being used for colorectal cancer treatment; however, the available drugs are still limited. Troster et al. reported the potential of EphA2 inhibition as a novel molecular-targeted therapy for colorectal cancer [29]. Feng et al. reported that targeting EphA2 and promoting its degradation inhibited cell proliferation in gastric and colorectal cancer in both in vitro studies and mouse models [30]. As the significance of EphA2 in colorectal has been elucidated, it is expected that EphA2 will become a useful biomarker for the early detection and personalized treatment of colorectal cancer.

The present study was associated with several limitations including its single-center setting and relatively small sample size. It is essential to accumulate results from a larger number of cases to enhance the reliability of the findings.

## 5. Conclusions

Serum EphA2 levels were significantly higher in patients with CRC than in healthy individuals, suggesting that it may be a biomarker that increases in the early stages. The measurement of serum EphA2 in patients with CRC is expected to be a practical, minimally invasive screening test for early detection and potentially more effective than conventional tumor markers.

## Figures and Tables

**Figure 1 biomolecules-14-01504-f001:**
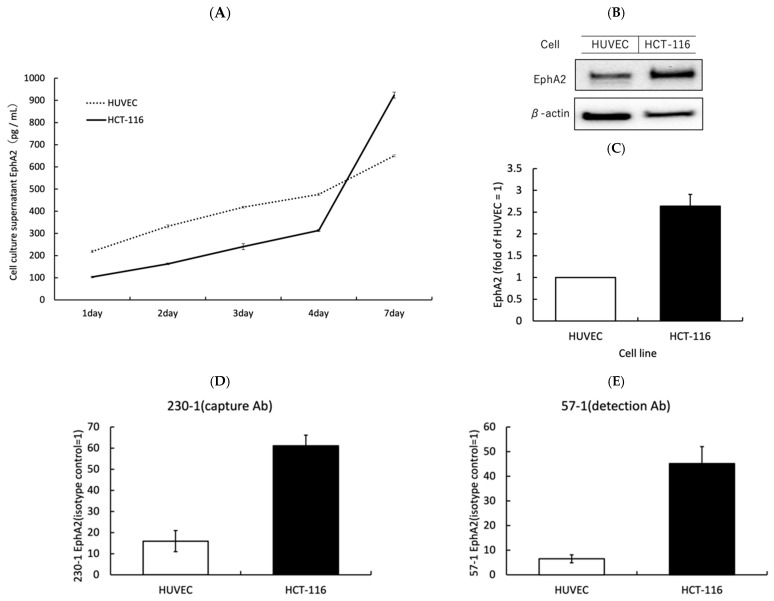
The analysis of extracellular release, and the intracellular and cell surface expression of EphA2: (**A**) EphA2 in cell culture supernatants was measured by an ELISA at each incubation time. (**B**,**C**) Western blotting was used to measure and quantify the total EphA2 content of the cells. (**D**,**E**), The expression of EphA2 on the cell surface was analyzed by flow cytometry using ELISA capture and detection antibodies.

**Figure 2 biomolecules-14-01504-f002:**
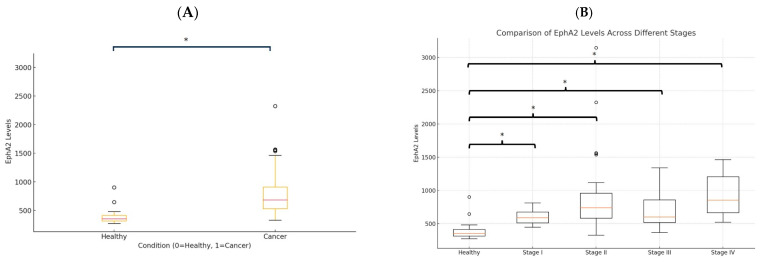
Soluble EphA2 levels in healthy individuals and colorectal cancer (CRC) patients. (**A**) EphA2 levels in healthy individuals and overall cancer patients (Stages I–IV). (**B**) EphA2 levels in healthy individuals and cancer patients according to stage. Circles represent individual data points, and asterisks indicate statistically significant differences (*p* < 0.05).

**Figure 3 biomolecules-14-01504-f003:**
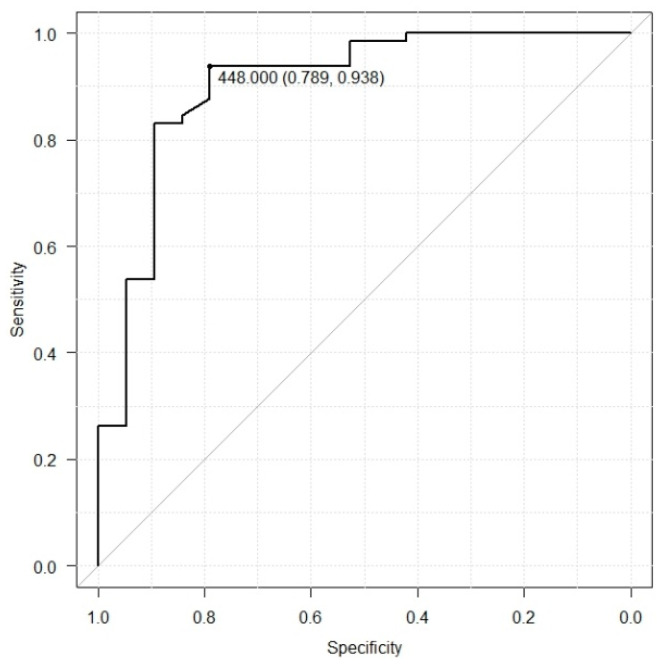
The ROC curve analysis of serum EphA2 levels in healthy individuals and colorectal cancer patients (AUC = 0.9, sensitivity = 93.8, specificity = 78.9, cutoff value = 448).

**Table 1 biomolecules-14-01504-t001:** Patient demographics and clinical characteristics.

Category	CRC *n* = 65
Age, Median (IQR)	73 (66–79)
Sex Male:Female	42:23
Location Right:Left	25:40
Stage I:II:III:IV	10:30:19:6

**Table 2 biomolecules-14-01504-t002:** Serum EphA2 in CRC patients based on clinical characteristics. “N.S.”: Not Significant.

Category		CRC (*n* = 65)	EphA2 (pg/mL) (Range)	*p* Value
Tumor location	Right	25	641 (329–3145)	N.S.
	Left	40	692 (359–2325)	
Tumor size	≤30 mm	18	636 (329–1117)	0.0346
	>30 mm	46	730 (359–3145)	
Depth of invasion	pT1–2	12	540 (449–812)	0.0311
	pT3–4	53	714 (329–3145)	
Lymph node metastasis	Negative	43	689 (329–3145)	N.S.
	Positive	22	607 (368–1465)	
Distant metastasis	Negative	59	672 (329–3145)	N.S.
	Positive	6	856 (522–1465)	
Vascular invasion	Negative	14	657 (449–2325)	N.S.
	Positive	48	689 (329–3145)	
Lymphatic invasion	Negative	13	812 (449–2325)	0.0431
	Positive	49	631 (329–3145)	
Stage	I–II	40	686 (329–3145)	N.S.
	III–IV	25	656 (368–1465)	
CEA (ng/mL)	≤5	39	642 (329–1565)	N.S.
	>5	25	782 (359–3145)	
CA19–9 (U/mL)	≤37	54	649 (329–3145)	N.S.
	>37	10	712 (359–1465)	

## Data Availability

The data presented in this study are available on request from the corresponding author due to privacy and ethical reasons.

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
