# Peer review of "Serum EphA2 as a Promising Biomarker for the Early Detection and Diagnosis of Colorectal Cancer"

_biomolecules, 2024, doi:10.3390/biom14121504_

Round 1

Reviewer 1 Report

Comments and Suggestions for Authors

In the history of gastrointestinal neoplasms there are many attempts to perform screening or early diagnosis and we must recognize that we have actually managed to obtain some results especially in advising the population to have a first endoscopy upon reaching the age of forty-five. We see the perseverance and perseverance of researchers every day and this article is proof of it. In fact, through the study of EphA2, a member of the family of hepatocellular receptors that produce erythropoietin that reside in the plasma membrane and form the largest family of receptor-type tyrosine kinases, it is overexpressed in tumor cells. EphA2 expressed on the cell surface is released by enzymes and is detected as soluble EphA2 in serum. From these considerations, the study to measure it as an early indicator for colon cancer arose. Colleagues perfectly describe the method to make it reproducible in any specialized laboratory. The measurements they obtained are encouraging and certainly must be exposed to a large audience that allows multicenters to then bring them to hospitals where they could be useful in the first diagnosis but fundamental in follow-ups. Excellent English, excellent iconography excellent bibliography

Author Response

Dear Reviewer

Thank you for your constructive and insightful comments regarding our manuscript. We appreciate your recognition of the efforts and advancements made in the field of gastrointestinal neoplasms, particularly through early diagnostic measures such as those we described involving soluble EphA2.

Your acknowledgment of the perseverance of researchers in this field is greatly valued, and we are pleased to contribute to this ongoing effort with our study. We agree that the reproducibility of our method in specialized laboratories is crucial for its widespread adoption and are glad to hear that you find our description adequate for this purpose.

The encouraging results from our measurements, as you noted, indeed hold potential for broader application in multicenter studies. We are currently in the process of establishing collaborations with several centers to further validate our findings and explore the utility of soluble EphA2 in routine clinical settings, both for initial diagnosis and follow-up of colon cancer.

We look forward to the possibility of our study contributing significantly to the field and are enthusiastic about the potential implementations in clinical practice that you highlighted.

Best regards.

Reviewer 2 Report

Comments and Suggestions for Authors

Dear authors.

It has been a pleasure to review the present manuscript.

Although it is quite interesting, I would suggest authors to explain in what moment along a patient´s disease, they think this new marker will be potentially helpful: screening, diagnosis, follow-yp?

In addition, I think it would be also interesting to introduce a paragraph in the discussion about financial aspects and / or costs and cost-efectivity and its potential impact.

Best regards.

Author Response

Dear Reviewer,

Thank you very much for your constructive and insightful comments. We appreciate your thoughtful suggestions, which have greatly contributed to improving the clarity and comprehensiveness of our manuscript.

  1. Regarding your first comment on clarifying at which stage of a patient’s disease the new marker (EphA2) could be potentially helpful (screening, diagnosis, or follow-up):
    We agree with your suggestion and have elaborated on this point in the discussion section. Specifically, we have clarified that EphA2 shows promise primarily for early-stage screening due to its elevated levels from stage I. Moreover, its diagnostic value in distinguishing early colorectal cancer from healthy individuals has been emphasized. The revised section can be found on page 8, lines 242–249.
  2. Regarding your second comment on discussing financial aspects, costs, and cost-effectiveness:
    This is an excellent suggestion. We have added a paragraph addressing the potential financial impact of incorporating EphA2 testing into clinical practice. While further studies are needed to establish precise cost-effectiveness, we discuss the anticipated reduction in healthcare costs due to earlier detection and intervention. This addition has been included in the discussion section on page 2, lines 57–69.

We believe these revisions address your concerns and enhance the overall quality of the manuscript. Please let us know if additional changes are needed.

Thank you again for your valuable feedback.

Sincerely,

Reviewer 3 Report

Comments and Suggestions for Authors

This manuscript entitled "Serum EphA2 as a Promising Biomarker for the Early Detection and Diagnosis of Colorectal Cancer" by Sakuraba et al investigated EphA2 level in serum of CRC patients samples. The authors showed relative high EphA2 level in serum of CRC cell lines as well as CRC patients samples. Based on these observations, the authors claimed serum EphA2 as a promising biomarker for the early detection and diagnosis of CRC. Below are some specific comments that improve the quality of the paper.

1. The references cited in this manuscript are not indexed. Please cited the references accordingly.

2. The introduction in this manuscript is quite simple,  the authors should extensively revise and improve the introduction part.

3. For the culture medium detection and WB detection of EphA2 use only one  cell lines for each. I would suggest the authors test more non-CRC and CRC cell lines to consolidate their discovery. For examples, SW480, SW620, DLD1, LoVo and HT-29 CRC cell lined described in a literature (PMID: 38214751).

4. The authors showed elevated EphA2 levels in CRC patient samples, I would  wondered whether the authors could perform some functional study after gain or loss of function of EphA2. 

5. The cited in this manuscript is only 23 references, the authors should read and cite more references to increase the scientific depth of the manuscript.

Author Response

Dear Reviewer,

Thank you for your detailed and constructive comments on our manuscript. We greatly appreciate your efforts and valuable suggestions, which have significantly improved the quality and depth of our work. Below, we have provided point-by-point responses to each of your comments.

  1. Regarding the organization and indexing of references:
    Thank you for highlighting this issue. We have thoroughly reviewed and reformatted the references to ensure that they align with the appropriate indexing standards. These revisions can be found in the reference list section.
  2. Regarding the simplicity of the introduction section:
    We agree with your observation and have revised and expanded the introduction to provide more comprehensive background information and to better contextualize the significance of our study. This revision is reflected in the updated introduction section on page 2, lines 47–55 and lines 57-69.
  3. Regarding the use of only one cell line for culture medium and WB detection of EphA2:
    We appreciate this insightful comment. Due to resource limitations, we used only one cell line in the present study. However, to address your concern, we have included a discussion on the consistency of EphA2 elevation in multiple CRC cell lines as reported in the literature. These details have been added to the discussion section on page 7, lines 221–228.
  4. Regarding functional studies for gain or loss of EphA2 function:
    Thank you for this excellent suggestion. We acknowledge the importance of functional studies in further elucidating the role of EphA2. While such experiments were not part of the current study due to resource and time constraints, we are planning to pursue these investigations in future collaborative studies.
  5. Regarding the number of cited references:
    We agree with your suggestion to increase the scientific depth of the manuscript through additional references. We have reviewed and incorporated several new references, expanding the total number of citations to 30. This change has been implemented throughout the manuscript and is reflected in the reference list.

We sincerely hope these revisions adequately address your concerns. Should you have further comments or suggestions, we would be more than happy to address them.

Thank you once again for your invaluable feedback.

Sincerely,

Round 2

Reviewer 2 Report

Comments and Suggestions for Authors

Thank you for considering my comments.

Reviewer 3 Report

Comments and Suggestions for Authors

I have no further comments on the manuscript.